# Enormous-stiffness-changing polymer networks by glass transition mediated microphase separation

Lie Chen[1,2], Cong Zhao[1], Jin Huang[1], Jiajia Zhou ⬤[1] ✉ & Mingjie Liu ⬤[1,3] ✉

The rapid development of flexible electronics and soft robotics has an urgent demand for materials with wide-range switchable stiffness. Here, we report a polymer network that can isochorically and reversibly switch between soft ionogel and rigid plastic accompanied by a gigantic stiffness change from about 600 Pa to 85 MPa. This transition is realized by introducing polymer vitrification to regulate the liquid–liquid phase separation, namely the Berghmans' point in the phase diagram of binary gel systems. Regulating the Lewis acid-base interactions between polymer and ionic liquids, the stiffness-changing ratio of polymer network can be tuned from 10 to more than $10^5$. These wide-range stiffness-changing ionogels show excellent shape adaptability and reconfigurability, which can enhance the interfacial adhesion between ionogel and electrode by an order of magnitude and reduce interfacial impedance by 75%.

Polymer materials with stiffness-changing property are advantageous for technological applications such as soft robotics, adhesives, and aeronautics because this mechanical duality offers a solution to the engineering paradox of shape adaptability and load-bearing capability[1–3]. Generally, thermoplastics are commonly used materials possessing a well-defined transition between hard and soft states upon crystallization-melting transition or glass transition[4]. These single-component polymer networks usually constrained by limited stiffness-changing ratio ($E_{hard}/E_{soft} \approx 10^3$) due to chain entanglement[5,6]. Forming binary systems by introducing liquid–solid phase change materials increases the degree of freedom in stiffness regulation of polymer networks. For example, combined with low melting point alloy, hydrated salt or crystalline polymer[7–9], the stiffness-changing ratio of the polymer composites can reach $10^4$–$10^5$. While, these binary systems depend solely on the intrinsic properties of phase change components, and thus suffer from limited tunability.

Liquid–liquid phase separation is a special form of a phase transition, which can lead to a decomposition of polymer gels into dense and sparse polymer regions while yielding stiffness change[10,11]. However, conventional polymer gels usually undergo weak phase separation with uncontrollable metastable states, accompanied by either relatively small modulus change (≈10-folds at most) or significant volume change[12–14]. Herein, we solve this problem by introducing polymer vitrification to regulate the liquid–liquid phase separation, namely the Berghmans' point in the phase diagram of binary gel systems. The resultant polymer networks can isochorically and reversibly switch between soft ionogels and rigid plastics with more than $10^5$-folds stiffness change (0.62 kPa versus 85.4 MPa). The vitrification of polymer-dense domain locks the metastable state during phase separation, resulting in a stable bicontinuous structures consisting of a vitrified phase and a gelated phase in polymer network. We found the Lewis basicity of ionic liquids (ILs) has significant influences on the Berghmans' point of corresponding gel systems. Varying the combination of cations or anions in ionic liquid blends, the stiffness-changing ratio of polymer network can be tuned from 10 to more than $10^5$. Taking advantages of the microstructure adaptability and reconfigurability of these materials, the interfacial adhesion and the interfacial impedance between ionogel electrolyte and Cu electrode are enhanced by an order of magnitude and reduced by 75%, respectively. These nonvolatile ionogels with wide-range switchable

[1]Key Laboratory of Bio-Inspired Smart Interfacial Science and Technology of Ministry of Education, School of Chemistry, Beihang University, Beijing 100191, China. [2]Nerve-Machine Integration and Cognitive Competition Center, Beijing Machine and Equipment institute, Beijing 100854, China. [3]International Research Institute for Multidisciplinary Science, Beihang University, Beijing 100191, China. ✉e-mail: jjzhou@buaa.edu.cn; liumj@buaa.edu.cn

thermomechanical properties may find potential uses in soft robotics, wearable devices and aeronautics.

The Berghmans' point is known as the intersection of binodal curve and the glass transition curve ($T_g$ curve) in the phase diagram of binary polymer solution. It was first proposed by Berghmans and Arnauts in 1987[15,16]. A representative feature for this system is the liquid–liquid demixing intercepted by the glass transition of the polymers. As shown in Fig. 1a, along the $T_g$ curve where Berghmans' point is reached, the glass transition temperature of the polymer-dense phase after phase separation becomes invariant with respect to the initial concentrations of the system. Early research had utilized this liquid–liquid phase separation coupled with polymer vitrification to "freeze" the temporary structure and morphology in polymer solution[17–20], but the implementation of Berghmans' point in designing functional materials is rare. It might be due to the fact that the Berghmans' point is not commonly observed; it only exists in binary polymer-solution system displaying an upper critical solution temperature (UCST) in addition a high $T_g$ of the used polymer. The difficulty to control its position in the phase diagram also limits the application and development of the Berghmans' point.

Herein, we extend the Berghmans' point to binary gel systems to prepare polymer materials with desirable mechanical duality. As shown in Fig. 1b, when the system is quenched to the phase-separation region above the Berghmans' point ($T > T_B$), the polymer gel decomposes into gelated-polymer-sparse and gelated-polymer-dense phases, accompanied by a rubbery-to-rubbery transition (Fig. 1b left). In contrast, when the system is quenched to the phase-separation region below the Berghmans' point ($T < T_B$), the polymer gel decomposes into a gelated-polymer-sparse phase and a vitrified-polymer-dense phase, accompanied by a rubbery-to-glassy transition (Fig. 1b right). For the latter case, the freeze of polymer-dense phase will simultaneously locks the metastable state during phase separation, thus restricts the large-scale phase separation. The vitrification of polymer-dense phase not only freezes the morphologically metastable state, but also arrests the compositional change ($\Phi_B$ in Fig. 1a) during the phase separation. As a result, the phase-separated polymer network form stable bicontinuous microstructures with fixed composition in each phase. This microstructure in return traps solvents inside the polymer networks, so the polymer networks exhibits isochoric and reversible switch between soft ionogels and rigid plastics with enormous stiffness change, representing an advance over conventional polymer materials.

## Results
### Stiffness-changing properties of PNIPAm ionogels
To construct a liquid–liquid phase separation displaying an UCST, we select poly N-isopropylacrylamide (PNIPAm) as a target polymer

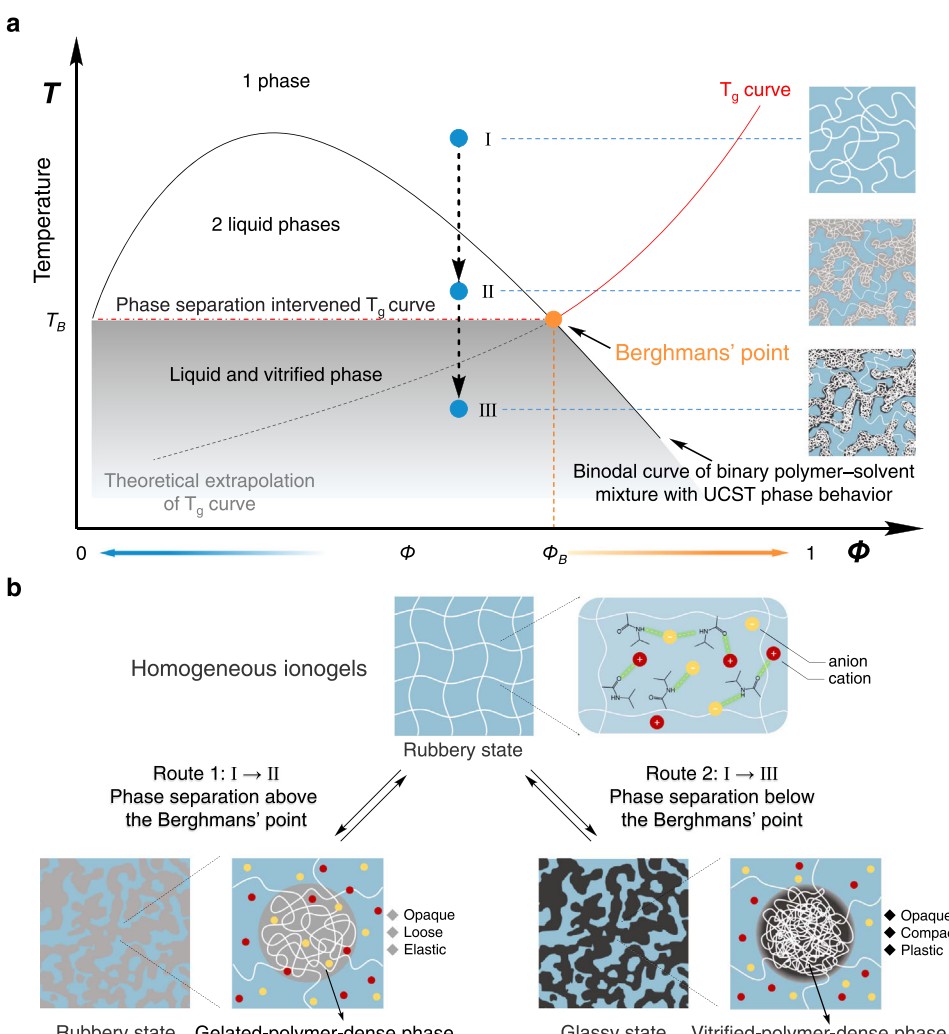

**Fig. 1 | Stiffness-changing mechanism of polymer networks. a** An illustrative phase diagram of solvent-polymer demixing interferes with the glass transition of the polymer. The broken line is the theoretical extrapolation of the glass transition line. At lower polymer concentrations ($\Phi < \Phi_B$), a concentration-independent glass transition temperature ($T_g = T_B$) is expected as presented by dot-dashed line. **b** Evolution of structures and states of ionogels quenching at different temperatures. Difference in polymer-dense phases are depicted when the system quenched to the phase-separation region above (left) and below (right) the Berghmans' point.

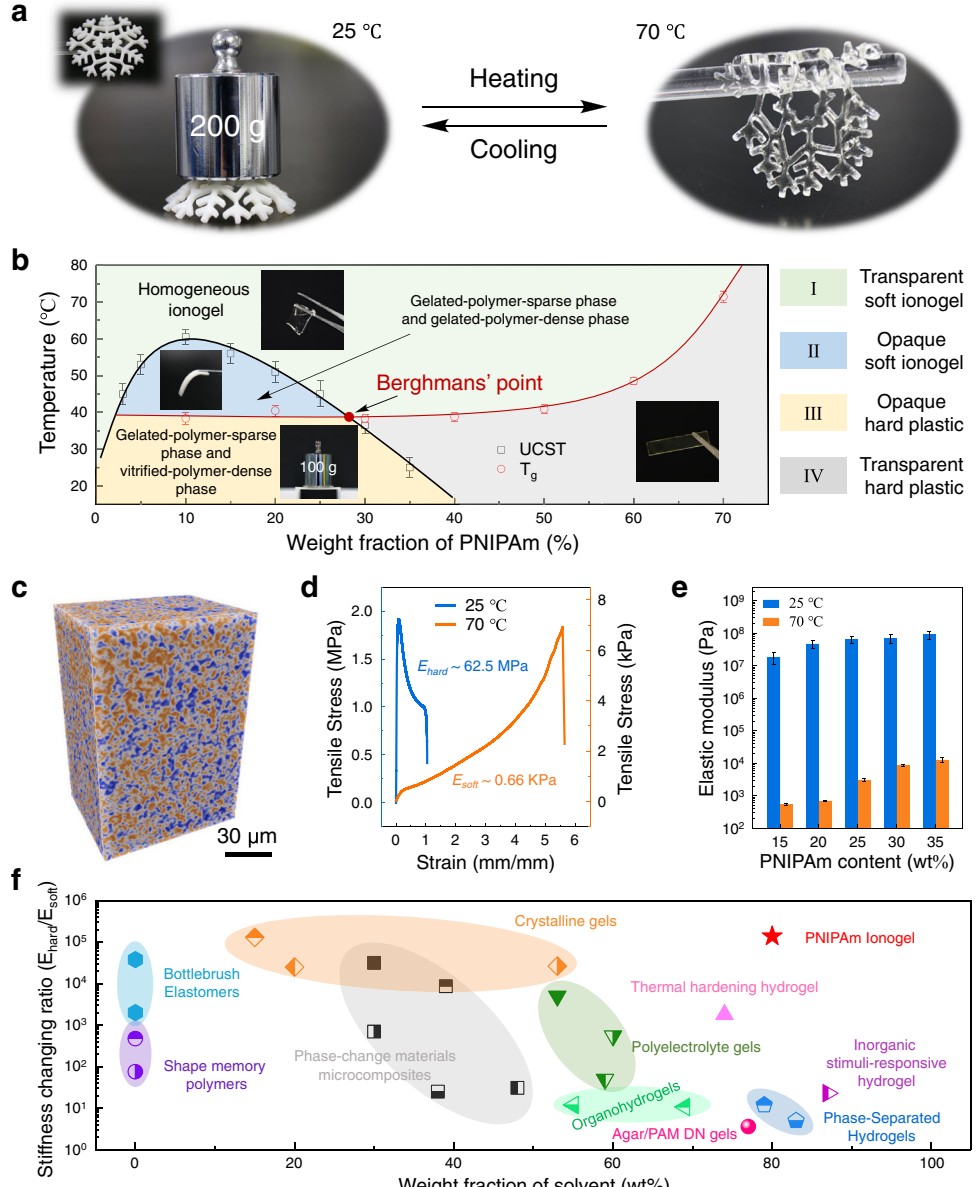

**Fig. 2 | Phase diagram and mechanical behaviors of [C₁MIM][NTf₂]-PNIPAm ionogels. a** The photographs illustrate the mechanical behavior of ionogel, which is stiff enough to hold a weight of 200 g in the rigid state but soft enough to stick to the glass bar in the soft state. **b** Phase diagram of PNIPAm ionogel composed of phase separation curve and the $T_g$ curve. **c** Nano CT image qualitatively illustrate the bicontinuous structure of the phase-separated ionogel (the blue phase represents the polymer-rich region and the yellow phase represents the ILs-rich region). **d** Stress−strain curves of PNIPAm ionogel under uniaxial tensile test at different temperatures. **e** Mechanical characterization of ionogels with different compositions using effective elastic modulus extracted from the initial stress−strain curves. **f** The stiffness-changing ratio of PNIPAm ionogel compared with other polymer-based stiffness-changing materials/devices with respect to the solvent content (or inactive component) (Supplementary Table 1). All error bars represent the mean ± standard deviations ($n \geq 3$ independent experiments).

network benefit from its flexibility in design liquid−liquid phase separation in binary system[21]. Herein, the PNIPAm ionogels with UCST phase behavior are prepared by free radical polymerization from NIPAm dissolved ILs solution (Supplementary Fig. 1). Unless specified, the behavior of the PNIPAm (20 wt%) ionogel using 1-Methyl-3-methylimidazolium bis(trifluoromethylsulfonyl)imide ([C₁MIM][NTf₂]) as solvents is described in detail as a representative sample. As shown in Fig. 2a, the as-prepared ionogel is in a white, plastic-like rigid state at room temperature. A temporary shape of convex upward snowflake ionogel can bear a load (200 g) more than 110 times of its own weight. When heated above the UCST, the ionogel became transparent and soft enough to stick to the glass bar. These suggest that PNIPAm ionogels exhibit remarkable stiffness changes along with their UCST-type phase transition. Correlating the optical

transmittance with the dynamic mechanical behaviors of PNIPAm ionogel during phase separation (Supplementary Fig. 2), we found that the glass transition of polymer-dense domain generated from phase separation is the key for this phenomenon. To confirm the thermodynamic behaviors of ionogel during phase separation, the relationship between the UCST and the $T_g$ of ionogels are investigated (Supplementary Fig. 3 and Fig. 4). We construct the phase diagram by plotting the UCST of ionogels as a function of PNIPAm content (Fig. 2b). The phase separation curve of ionogel is convex downward, resembling to the typical UCST phase diagram of a binary mixture. The critical temperatures is about 60.5 °C with the critical composition of 10 wt%. For comparative analysis, we add the $T_g$ curve to the phase diagram as shown in Fig. 2b. The intersection of the $T_g$ curve and the phase separation curve is the Berghmans' point as indicated by the red

dot. Phase separation above the Berghmans' point (region II) results in a gelated-polymer-sparse phase and a gelated-polymer-dense phase in a rubbery state. While, phase separation below the Berghmans' point (region III), the polymer-dense phase vitrifies before it has reached its equilibrium composition defined by the phase separation curve. In this system, the temperature at Berghmans' point turns out to be about 40 °C. Therefore, when the phase-separated ionogel (e.g. 20 wt% PNIPAm content) quenched to room temperature, the polymer-dense phase vitrifies and stiffen the ionogel, resulting in an ionogel with high mechanical performance. It is worth noting that the temperature-transmittance curves have a strong cooling rate dependence when UCST is close or lower than $T_g$, which can be utilized to prepare ionogel with special optical and mechanical properties (Supplementary Fig 5). The internal structure of phase-separated ionogel can be visualized via the Nano Computed Tomography (Nano CT) scanning (Fig. 2c). Similar to traditional microphase separation, the vitrified-polymer-dense phase and gelated-polymer-sparse phase form bicontinuous structures of micrometer scale[22,23]. Further, we investigate the solvent trapping capability of this bicontinuous microstructure using 25 wt% PNIPAm/[C_1MIM][NTf_2] ionogel (UCST ~ 45°C) as an example. Results indicates this bicontinuous microstructure can trap 90.7% solvent inside the phase-separated ionogel when held at room temperature for 4 days (Supplementary Fig 6).

We then characterize the mechanical behaviors of PNIPAm ionogels through a uniaxial tensile test. As shown in Fig. 2d, below the UCST, the stress–strain curve of PNIPAm ionogel displayed a plastic deformation. The plastic ionogel is hard but not brittle, which can achieve 100% deformation with a typical plastic yielding strain like thermoplastics, while keeping 80 wt% of solvent content. In contrast, the ionogel exhibited elastic deformation with an elongation at break of ~560% at 70 °C. The changes in plastic/elastic behavior are accompanied by an enormous change in elastic modulus (62.5 MPa versus 0.66 kPa), suggesting a significant change in stiffness between two states. This enormous stiffness change is attributed to the coexistence of vitrified-polymer-dense phase and gelated-polymer-sparse phase in the system. The continuous-vitrified phase provides high mechanical strength for phase-separated polymer gel. While, the gelated-polymer-sparse phase ensure that the polymer network are fully solvated (i.e. low modulus) when the system become homogeneous. Besides, the influence of PNIPAm content on the stiffness-changing property of ionogel are characterized and the results are given in Fig. 2e (stress–strain curves are provided in Supplementary Fig. 7). The maximum stiffness-changing ratio of ionogel at two states is generated from 20 wt% of PNIPAm content. At low polymer concentrations ($\Phi < 35$ wt%), the elastic modulus are increased for ionogels at each state as increasing PNIPAm content. In homogeneous state, the modulus increase (from 0.54 kPa to 12.7 kPa) is ascribe to the increase in polymer network density. In phase-separated state, the key reason for elastic modulus enhancement (from 18.2 MPa to 92.2 MPa) is the volume increase of vitrified-polymer-dense phase, which is qualitatively demonstrated by the Nano CT results (Supplementary Fig. 8). Because in this system, the composition (i.e. density) of vitrified-polymer-dense phase ($\Phi_B$) has been determined by the Berghmans' point (Fig. 2b). From the above discussion, we can conclude that the Berghmans' point regulates the mechanical properties of phase-separated ionogel from two aspects: the state (Y-axis, i.e. temperature) and composition (X-axis) of the polymer-dense phase. Further, to demonstrate the superiority of this strategy, the stiffness-changing property of PNIPAm ionogel is compared with other polymer-based stiffness-changing materials by plotting the maximum stiffness-changing ratio against their solvent (or inactive component) content (Fig. 2f). (The maximum elastic modulus change of PNIPAm ionogel is generated from [C_5MIM][PF_6]-PNIPAm ionogel with 20 wt% PNIPAm content, 85.4 MPa versus 0.62 kPa).

## Regulation of the Berghmans' point

To gain further insight into this observation, we investigate the structural effects of ILs on the UCST and the Berghmans' point of PNIPAm ionogels by varying the combination of cations and anions in ILs (Fig. 3a shows the structural formula of the cations and anions used in this work). The influences of cations and anions on the UCST of PNIPAm ionogels are illustrated in Fig. 3b. For a certain anion, the UCST of ionogels decreased as increasing the cation side chain length. While, for a certain cation, there is no clear rule for the effect of anions on the UCST of ionogels. To explain this, understanding the influence of cation and anion on ILs is prerequisite. It has been demonstrated before that the polarity of ILs appears to be largely cation controlled, while the Lewis basicity (or donor strength) is mainly anion dependent[24,25]. Consequently, the influence of cations on the UCST of ionogels is due to the polarity change of their corresponding ILs. Increasing the alkyl substituent length on cation reduces the polarity of ILs. ILs with lower polarity has better interactions with the nonpolar isopropyl groups on PNIPAm, resulting in a lower UCST of the ionogel. In comparison, the influence of anions on the UCST of ionogels are much more complicated. As shown in Fig. 3b, for certain cation [C_4MIM], the UCST of PNIPAm ionogels follow the order [OTf] < [NTf_2] < [PF_6] < [BF_4]. However, the Lewis basicity of ILs ([PF_6] < [NTf_2] < [BF_4] < [OTf]), which mainly determines the mutual solubility between polymer and ILs do not shows any correlation with the UCST of corresponding ionogels[26,27]. We hypothesize that the factor determines the UCST of PNIPAm ionogels is the overall properties of ILs, rather than a certain property that generated from the cation or anion in ILs[28].

The effects of cation and anion on the mechanical property of PNIPAm ionogels are characterized by comparing the elastic modulus of PNIPAm ionogels in different ILs solvents (Mechanical properties of three representative samples are summarized and provided in Supplementary Table 2). As shown in Fig. 3c, for ionogels using ILs with certain anion ([NTf_2]), the change of cation ([C_1MIM], [C_4MMIM] and [C_3MPY]) hardly affects the tensile modulus of the phase-separated ionogels ($10^7$-$10^8$ Pa). While for ionogels using ILs with certain cation ([C_5MIM]), the change of anion shows great effect on the tensile modulus of phase-separated ionogels ($10^7$-$10^8$ Pa for [PF_6] and $10^3$-$10^4$ Pa for [BF_4]). We supposed that the anionic Lewis basicity is the key to explain these results, because PNIPAm is easily soluble in a solvent having relatively high Lewis basicity by forming hydrogen bonds[29]. When phase separation occurs in ionogel using ILs with high Lewis basicity anions ([BF_4] and [OTf]), the ILs is difficult to exclude from the polymer-dense phase, leading to a relative weak phase separation with slow kinetics. In contrast, PNIPAm is not soluble in a solvent with low Lewis basicity ([PF_6] and [NTf_2]), thus the ILs will easily exclude from the polymer-dense phase with a fast kinetics when phase separation occurs. This hypothesis is also confirmed by the phase separation curves in Fig. 3e. Although the anion species has significant influences on the phase separation kinetics of ionogels, the key factor that determines the mechanical properties of phase-separated iono-gels lies in the Berghmans' point in the phase diagram. Therefore, the effects of cation and anion on phase diagram and the Berghmans' point of corresponding ionogel system needs to be clarified.

Under the guidance of the results in Fig. 3c, we construct the phase diagrams and corresponding $T_g$ curves of four ionogel systems using ILs with different combinations of cation and anion (DSC results are provided in Supplementary Fig. 9). As shown in Fig. 3d, the influence of cation on phase diagram is the upward or downward translation of the phase separation curve without changing the critical compositions. In comparison, the anion significantly affects the phase separation curve by changing the critical temperatures as well the critical compositions in phase diagram (Fig. 3e). Besides, it is found that the wider the phase separation curve opens, the higher it intersects with the $T_g$ curves. The difference in the opening size of phase

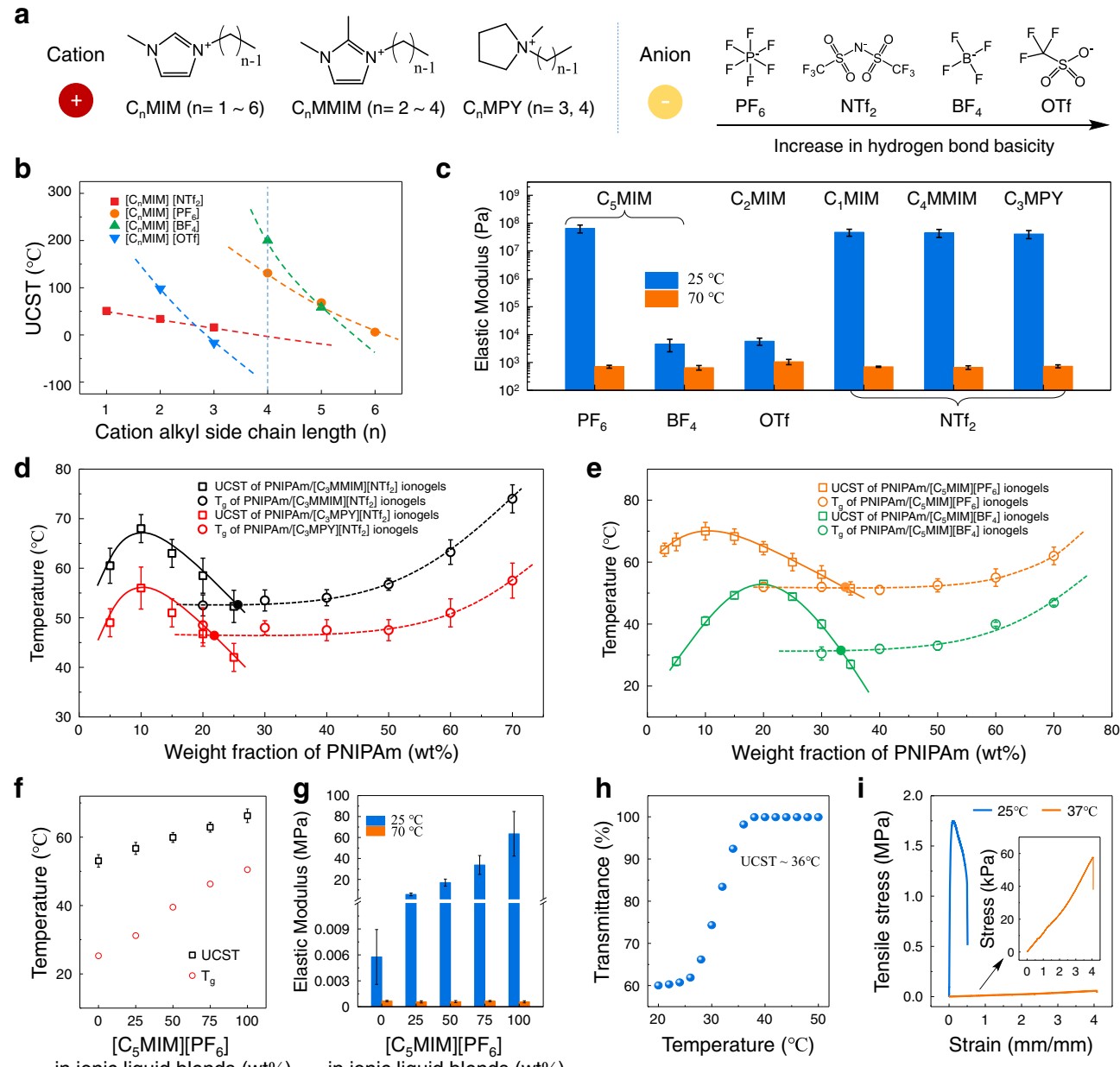

**Fig. 3 | Structural effects of ILs on the Berghmans' point. a** Molecular structures of cations and anions of ILs used in this work. **b** UCST of ionogels with respect to the molecular structure ILs. **c** Mechanical characterization of PNIPAm ionogels using different ILs as solvent with elastic modulus extracted from the initial stress−strain curves (the UCST of ionogels from left to right are 66.3 °C, 53.2 °C, 98 °C, 51 °C, 52 °C and 47 °C, respectively). **d** Effect of cation in ILs on the phase diagram of the ionogels. **e** Effect of anion in ILs on the phase diagram of the ionogels. **f** Tuning the UCST and $T_g$ of the ionogels by varying the mixing ratio of anions in ionic liquids blends ([C$_5$MIM][PF$_6$]/[C$_5$MIM][BF$_4$]). **g** Thermoresponsive stiffness-changing properties of the ionogels in **f**. **h** Temperature dependence of transmittance at 658 nm for PNIPAm ionogels with certain feed ratios of constituents (PNIPAm content: 32 wt%, ionic liquid blends: [C$_5$MIM][PF$_6$]:[C$_5$MIM][BF$_4$] = 1:3). **i** Tensile stress−strain curves of PNIPAm ionogels at 25 °C and 37 °C, respectively. The elastic modulus of ionogels at 25 °C and 37 °C are 50.2 MPa and 11.4 kPa, respectively. All error bars represent the mean ± standard deviations ($n \geq 3$ independent experiments).

separation curves in Fig. 3e may be generated from the difference in phase separation kinetics of PNIPAm ionogel. As mentioned above, the phase separation in [C$_5$MIM][PF$_6$]-ionogels possesses fast kinetics, meaning that the right part of the phase separation curve is relatively less steep, so that the Berghmans' point emerged at an elevated temperature. Conversely, for [C$_5$MIM][BF$_4$]-ionogels, the slow kinetics of phase separation results in a relatively steep phase separation curve and a lower Berghmans' point in the phase diagram. In a word, the main way that cations regulate the Berghmans' point is by changing the UCST of ionogels, while anions in addition can affect the Berghmans' point by regulating the phase separation kinetics of ionogels.

The applications of PNIPAm ionogels with fixed feed ratios of constituents in a given area may require a specific transition temperature or stiffness-changing range, and therefore, tuning of the phase separation temperature and the glass transition temperature of these ionogels is of great significance. Benefit from the unlimited design flexibility of ILs, we can continuously regulate the UCST as well the $T_g$ (i.e. Berghmans' point at low polymer content) of PNIPAm ionogels by varying the mixing ratio of two ILs in their blends without modifying the chemical structure of the polymers[30–32]. By varying the combination of cations in ionic liquid blends (Supplementary Fig. 10, [C$_3$MMIM][NTf$_2$]/[C$_3$MPY][NTf$_2$]), the UCST and $T_g$ of ionogels can be

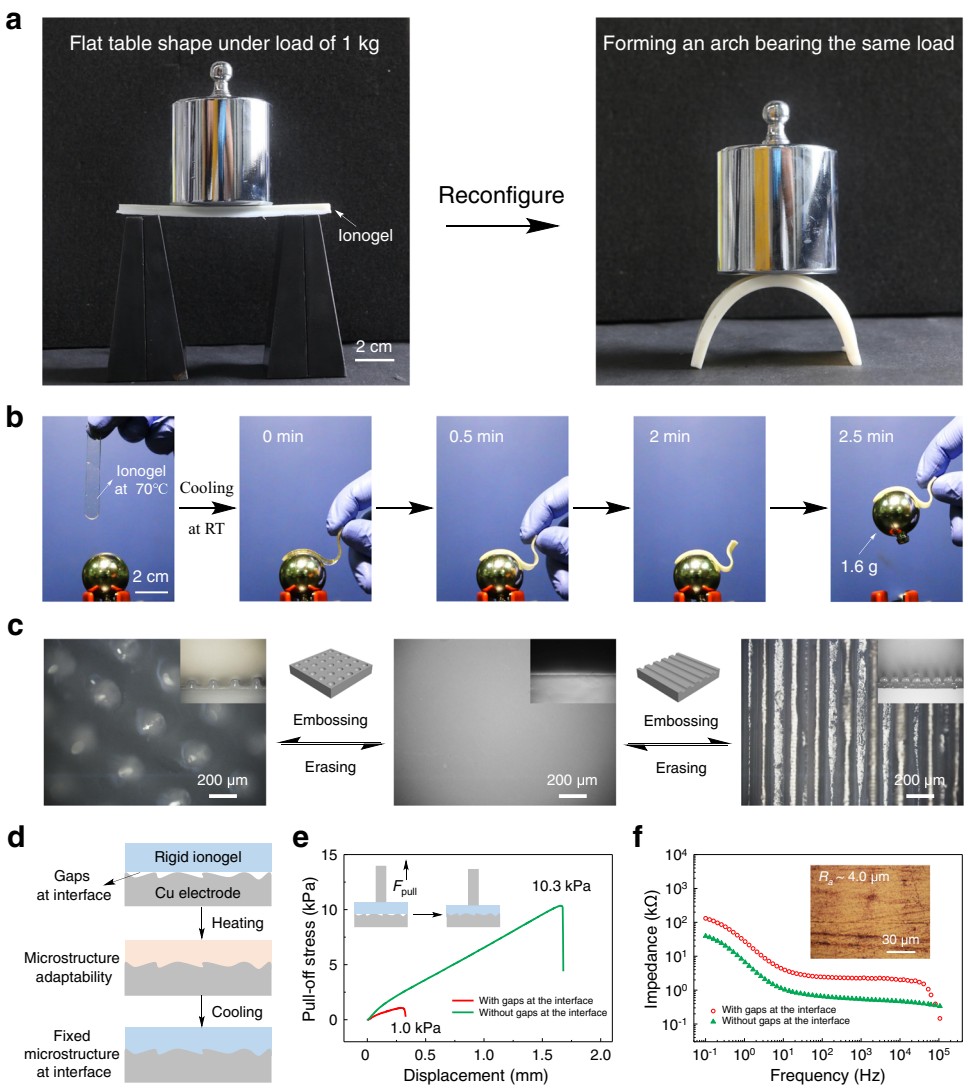

**Fig. 4 | Shape reconfigurability and microstructure adaptability of PNIPAm ionogels. a** Photographs showing the load-bearing capability and the shape reconfigurability of phase-separated PNIPAm ionogel. **b** Photographs showing the soft-to-hard transitional contact of PNIPAm ionogels. **c** Demonstration of the microstructure adaptability of PNIPAm ionogels. The semi-spherical pattern (left) and the ribbon pattern (right) were embossed by fixing the microstructure of substrate, which can be erased and recovered to the smooth original pattern (middle). **d** Schematic illustration of the microstructure adaptability of the ionogels. **e** The pull-off stress of ionogel on Cu electrode with Ra ≈ 4 µm. **f** Impedance of the ionogel/Cu system from 0.1 to $10^5$ Hz. As a result, the impedance of ionogel/Cu system at 1 Hz reduced from 26.9 kΩ to 6.6 kΩ by eliminating the gaps at the interface (the inset micrograph shows the surface topography of Cu electrode with a roughness of Ra ≈ 4 µm, Supplementary Fig. 16).

tuned to exhibit a linear variation from 47 °C to 58.7 °C and from 44.8 °C to 53.4 °C, respectively (DSC results are provided in Supplementary Fig. 11a). These ionogels exhibit high stiffness-changing ratio between two states ($10^7$-$10^8$ Pa versus $10^2$-$10^3$ Pa). The increase in modulus of phase-separated ionogels (from 39.6 MPa to 47.3 MPa) as increasing the proportion of [$C_3$MMIM][$NTf_2$] in ionic liquid blends is ascribe to the rightward shift of the Berghmans' point (i.e. $\Phi_B$) in the phase diagram (Fig. 3d). Moreover, by varying the combination of anions in ionic liquid blends as shown in Fig. 3f ([$C_5$MIM][$PF_6$]/[$C_5$MIM][$BF_4$]), the $T_g$ of ionogels can be tuned within a wider temperature range from 25.3 °C to 50.5 °C under a slightly change of corresponding UCST (DSC results are provided in Supplementary Fig. 11b). As a result, the as-prepared ionogels exhibit controllable stiffness-changing ratio from 10 to $10^5$ (stress–strain curves are provided in Supplementary Fig. 12) within a temperature range from 25 °C to 70 °C (Fig. 3g). For the applications of ionogels in a given area such as wearable devices, thermoresponsive ionogel within body temperature (25 °C-37 °C) can be prepared by regulating the feed ratios of constituents (Fig. 3h). The

stiffness of this ionogel can be switched between 50.2 MPa and 11.4 kPa (~4400-folds) within temperature range from 25 °C to 37 °C (Fig. 3i). It is worth noting that these ionogels also show unprecedented stiffness-changing ratio and temperature sensitivity that surpass all reported polymer-based stiffness-changing materials (Supplementary Table 3).

## Practical implications of ionogels

These nonvolatile ionogels with thermoresponsive ultra-wide stiffness-changing range exhibit original capabilities for applications. These include large-strain (900%) shape memory and mechanical energy storage (Supplementary Figs. 13 and 14). One major advantage towards the application of PNIPAm ionogels is that the shape reconfigurability and shape adaptability. We show this through manually configure the ionogel into a flat table (Fig. 4a left) and an arch (Fig. 4a right) and apply confining pressures. The phase-separated ionogels are mechanically stiff and able to bear mechanical loads, over 100 times their own weight. The shape adaptability is demonstrated through the soft-to-hard transitional contact of ionogel with a plastic ball (Fig. 4b).

In a homogeneous state, the soft ionogel can freely bend and drape over curved objects. As the shape of ionogel fixed after cooling at room temperature, the ionogel forms strong interaction with the ball, and the ball can be lifted along with the ionogel. The microstructure adaptability of PNIPAm ionogels is confirmed through the construction of micro-scale structures, such as semi-spherical pattern and ribbon pattern on the ionogel surfaces by embossing (Fig. 4c). On the basis of the microstructure adaptability, the ionogel can be utilized to eliminate the gaps at the interface between ionogel and substrates as illustrated in Fig. 4d. By eliminating the interfacial gaps, the adhesion strength between ionogel electrolytes and Cu electrodes increased by an order of magnitude from 1.0 kPa to 10.3 kPa (Fig. 4e). Meanwhile, the good compliance of ionogel increases the induced charge density at the interface and leads to low interfacial impedance with electrodes. As shown in Fig. 4f, the impedance of ionogel/Cu system reduced from 26.9 kΩ to 6.6 kΩ at 1 Hz by eliminating the gaps at the interface. Moreover, as an extended application base on the above-mentioned features, these PNIPAm ionogels also can be used as an intelligent gripper or in developing smart soft-matter machines, such as thermoresponsive pneumatic actuator (Supplementary Fig. 15).

## Discussion

In summary, we construct a liquid–liquid demixing intercepted by the glass transition of the polymer in binary gel system using PNIPAm and ILs, as well propose a universal strategy to regulate the mechanical properties of phase-separated ionogels base on the manipulation of Berghmans' point in the phase diagram. Through the combination of cations or anions in ionic liquid blends, the stiffness-changing ratio of PNIPAm ionogels can be tuned from 10 to more than $10^5$ under a mild temperature condition. This characteristic of nonvolatile ionogels enable their capabilities for applications in soft robotics, adhesives and aeronautics. Understanding the internal mechanism of current system in molecular scale allows us to extend this concept to other gel systems with different functionalities, such as thermal hardening, which substantially contribute to the development of functional polymer materials.

## Methods

### Preparation of ionogels

Generally, NIPAm (3 wt%, 5 wt%, 10 wt%, 15 wt%, 20 wt%, 25 wt%, 30 wt%, 40 wt%, 50 wt%, 60 wt%, 70 wt% were prepared), Ethyleneglycol dimethacrylate (crosslinker, 0.5 mol%, 1 mol%, 2 mol% and 5 mol% of monomer amount) and Diethoxyacetophenone (photo-initiator, 0.1 wt% to monomers) were added to ionic liquids. NIPAm was dissolved in ionic liquids by heating in an oven at 80 °C, then the mixture was stirred for 10 minutes until a homogeneous and fully transparent solution was obtained. The polymerization was initiated by UV irradiation using a mercury lamp (100 W) for 15 min. To prevent the crystallization of NIPAm monomers during the preparation of ionogel, the homogenous solution was kept at 40 °C when photo-induced polymerization.

### Mechanical characterization of ionogels

The stress–strain test of ionogels were executed using tensile machine (Electronic universal testing machine, SUNS, UMT4103). Ionogels were cut into dumbbell shape, 30 mm in length, 5 mm in width, 2 mm in thickness. Both ends of the dumbbell-shaped sample were connected to the clamps with the lower clamp fixed. The upper clamp was pulled by the load cell at a constant velocity of 5 mm/min at room temperature (approximately 24 °C), by which the stress–strain curve was recorded and the experimental data was further analyzed. The tensile strength was obtained from the failure point. The modulus was determined by the average slope over 0 ~ 20% of strain ratio detected from the stress–strain curve.

The adhesive force between ionogel electrolyte and Cu electrode (Fig. 4d) was obtained using the above-mentioned mechanical tester (SUNS, UMT4103) with 100 N load cell at a constant velocity of 5 mm/min under room temperature.

### Rheological test

The mechanical performance of ionogels were also characterized using Rheometer (Anton Paar, MCR 302). Ionogels with diameter of 15 mm and 2 mm thickness were prepared. Rheological test was carried out under room temperature with a strain of 0.5% and frequency of 1-100 rad/s. Temperature-sweep rheological test of PNIPAm ionogel (Fig. S2) was carried out at a heating rate of 5 °C/min (from 20 °C to 70 °C) with a strain of 0.5% and frequency of 1 Hz.

### Determination of cloud point

The cloud point (i.e. UCST) of ionogels were determined by optical transmittance measurements at 658 nm at a heating rate of roughly 1 °C/min using dynamic light scattering (Anton Paar, Litesizer™ 500). Ionogels were cut into rectangle shape (20 mm × 8 mm × 1.5 mm) and placed on the inner wall of cuvette. We define the cloud point values as the temperatures at which the transmittance reaches 90%.

### Thermal analysis of ionogels

The thermal analysis were conducted using dynamic thermo-mechanical analysis (DMA) (Q800, TA Instruments, New Castle, DE) and Differential scanning calorimeter (DSC) (Q2000, TA Instruments, New Castle, DE). The DMA experiments were conducted in a tensile mode, and the ionogels were cut into rectangle shape (15 mm in length, 5.6 mm in width, 0.85 mm in thickness). The sample was first annealed at 40 °C for 10 min prior to testing. The DMA curve (Fig. S11a) was obtained in a "DMA controlled force" mode.

For DSC test of ionogels, the sample was equilibrated at 25 °C, and then cooled to 0 °C using a cooling rate of 10 °C/min. The sample was held at 0 °C for 5 min, and then heated to 100 °C at a heating rate of 10 °C/min.

For DSC test of ionic liquids, 10 μL of ionic liquids was dropped into an aluminum hermetic pan, and then the pan was sealed. The sample was equilibrated at 40 °C, and then cooled to −5 °C using a cooling rate of 10 °C/min. The sample was held at −5 °C for 3 min, and then heated to 80 °C at a heating rate of 10 °C/min.

## Data availability

The data that support the findings of this study are available from the corresponding authors upon request.

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

## Acknowledgements

This work was financially supported by the National Key Research and Development Project (2022YFA1503000), National Natural Science Funds for Distinguished Young Scholar (No. 21725401), the National Natural Science Foundation (22161142021), and the China Postdoctoral Science Foundation (2019M650434).

## Author contributions

L.C. and M.L. contributed to the initial idea and designed the experiments. L.C. and C.Z. performed the experiments. L.C., J.H., J.Z. and M.L. analyzed the results. L.C., J.Z. and M.L. wrote the manuscript.

## Competing interests

The authors declare no competing interests.
