## [Peer Review File · Nature Communications]

Enormous-stiffness-changing polymer networks by glass transition mediated microphase separationReviewers' Comments:

Reviewer #1:

Remarks to the Author:

Materials with switchable stiffness are needed for a myriad of applications. Stiffness changing behavior is common in living organisms, serving as an inspiration for synthetic materials. Living organisms use stiffness-changing behavior to better adapt to various conditions. In synthetic materials, it is difficult to achieve reversible, extreme switchability of mechanical properties without premature failure. In this paper, the authors developed smart stiffness-changing materials, ionogels, that show excellent shape adaptability and reconfigurability. This serves as an elegant example demonstrating the development of reversible switch between soft ionogel and rigid plastic accompanied by a large stiffness change.

This is a well written, elegant paper describing efforts proposing universal strategy to regulate the mechanical properties of phase-separated ionogels base on the manipulation of Berghmans' point in the phase diagram. The experiments are well-designed, the results are discussed with insight. In addition, the conclusions are sound. The authors of the paper have clearly done a lot of work, and the results obtained by them are quite interesting. Overall, this study is of scientific and practical significance and should be published as is.

Reviewer #2:

Remarks to the Author:

The manuscript entitled "Enormous-stiffness-changing polymer networks by glass transition mediated microphase separation" by Chen et al. discusses how PNIPAm-based hydrogels in ionic liquids have different UCST and stiffness properties. Specifically, the authors described the effects of cations and anions on the phase transition of PNIPAm in ILs. They suggested that the Berghman point on the phase diagram is the key factor dictating the mechanical properties of thermoresponsive material during the phase transition. With these findings, the authors demonstrated a few cases where the design of PNIPAm-IL composite was successfully used in a few settings to hold or lift loads. The results are interesting and the approach offers a new route to fabricate new class of materials. However, clarifications on the theoretical part of the work is needed before consideration for publication. Details are below:

1. Lines 27-28: "Stiffness changing ratio" is reported with a unit. Revisit the reported values.
2. Lines 84-88: The key fundamental hypothesis in this work is that the microstructures trap solvents inside the polymer networks during the phase transition, allowing a reversible switch between soft ionogels and rigid plastics. Have the authors validated this hypothesis? For example, does the composite expel the solvent if held at $T > UCST$ for an extended period of time? To what degree the composite trap the solvent ("de-swollen" ratio)?
3. Lines 172-183: The authors discuss that the polarity of ions may not be enough to describe the behavior of ILs, and thus, suggest that the "overall properties" of ILs must be considered. While it is understandable the polarity may not be enough, but "overall properties" is still vague. Can the authors elaborate on what they mean by that term and how this needs to be considered for an IL? And, how do they support their suggestion?
4. Lines 271-272: Did the authors follow a specific protocol/procedure to "eliminate the interfacial gap"? was it a spontaneous step or the authors had to wait for a certain time to let the trapped gas escape?
5. Figure 1: what do "opaque loose elastic" black symbols show on the Figure 1b (left)? Are they

missing from the figure or they must be in grey?

6. Typos and grammatical errors throughout the manuscript. Examples: lines 32-34, line 63 ("research" is not countable), lines 184-186, line 189 ("effect" is a noun and "affect" is a verb), lines 213-216, lines 217-219, lines 246-247.

Reviewer #3:

Remarks to the Author:

In this paper, the authors developed a material that show rubbery-to-glassy transition upon cooling based on a binary polymer-solution system with an upper critical solution temperature (UCST) and a high T_g of the polymer. Similar rubbery-to-glassy transition upon heating based on a binary polymer-solution system displaying a lower critical solution temperature (LCST) with a high T_g of the polymer has been reported recently (Nonoyama et al., *Advanced Materials* 2020). Although this work reports a thermal softening while the previous report by Nonoyama et al is thermal stiffening, the underlying physical concept are the same, that is, using liquid-liquid phase separation of binary polymer-solution system and polymer vitrification to achieve large modulus switching, which largely weakens the novelty of this work. However, since few liquid-liquid phase separation systems have the binodal curve and the glass transition curve intersected (so called Berghmans' point) in the observation temperature window, this work using PNIPAm and ionic liquid is valuable in extending the material design range.

Specific comments:

- 1) The previous paper (Nonoyama et al, *Advanced Materials* 2020) should be properly cited.
- 2) Since the Berghmans' point freeze the kinetics, the temperature -transmittance curves (figure S3) should have a strong cooling rate dependence when UCST is close or lower than T_g . Therefore, the binary curve in Figure 2b and UCST curves in Figure 3b,d,e,f for the related data points should also strongly depend on the cooling rate. The authors should carefully address such cooling rate effect. I am wondering how the authors could determine the UCST when it is lower than T_g .

Responses to Reviewers' comments

Reviewer #1:

Comment 1. Materials with switchable stiffness are needed for a myriad of applications. Stiffness changing behavior is common in living organisms, serving as an inspiration for synthetic materials. Living organisms use stiffness-changing behavior to better adapt to various conditions. In synthetic materials, it is difficult to achieve reversible, extreme switchability of mechanical properties without premature failure. In this paper, the authors developed smart stiffness-changing materials, ionogels, that show excellent shape adaptability and reconfigurability. This serves as an elegant example demonstrating the development of reversible switch between soft ionogel and rigid plastic accompanied by a large stiffness change.

This is a well written, elegant paper describing efforts proposing universal strategy to regulate the mechanical properties of phase-separated ionogels base on the manipulation of Berghmans' point in the phase diagram. The experiments are well-designed, the results are discussed with insight. In addition, the conclusions are sound. The authors of the paper have clearly done a lot of work, and the results obtained by them are quite interesting. Overall, this study is of scientific and practical significance and should be published as is.

Response: We sincerely appreciate the Reviewer's time and effort for evaluating our manuscript as well as the very positive comments on our work.

Reviewer #2:

Comment 1. The manuscript entitled "Enormous-stiffness-changing polymer networks by glass transition mediated microphase separation" by Chen et al. discusses how PNIPAm-based hydrogels in ionic liquids have different UCST and stiffness properties. Specifically, the authors described the effects of cations and anions on the phase transition of PNIPAm in ILs. They suggested that the Berghmans' point on the phase diagram is the key factor dictating the mechanical properties of thermoresponsive material during the phase transition. With these findings, the authors demonstrated a few cases where the design of

PNIPAm-IL composite was successfully used in a few settings to hold or lift loads. The results are interesting and the approach offers a new route to fabricate new class of materials.

Response: Thanks for the Reviewer's positive comment.

However, clarifications on the theoretical part of the work is needed before consideration for publication. Details are below:

Comment 2. Lines 27-28: "Stiffness changing ratio" is reported with a unit. Revisit the reported values.

Response: We thank the Reviewer for this suggestion, the "stiffness-changing ratio" should not have a unit. This sentence has been revised as "stiffness-changing ratio ($E_{\text{hard}}/E_{\text{soft}} \approx 10^3$)" in the manuscript.

Comment 3. Lines 84-88: The key fundamental hypothesis in this work is that the microstructures trap solvents inside the polymer networks during the phase transition, allowing a reversible switch between soft ionogels and rigid plastics. Have the authors validated this hypothesis? For example, does the composite expel the solvent if held at **T > UCST** for an extended period of time? To what degree the composite trap the solvent ("de-swollen" ratio)?

Response: First of all, what reviewer concern should be "does the composite expel the solvent if held at **T < UCST** for an extended period of time?" Cause when above the UCST, the ionogel is in a homogeneous state. Follow the Reviewer's suggestion, we investigate the solvent trapping capability of bicontinuous microstructure. Taking 25 wt% PNIPAm/[C₁MIM][NTf₂] ionogel (UCST ~ 45°C) as an example, a time dependent mass variation of phase-separated ionogel at room temperature was recorded. For current sample, only less than 7 wt% solvent was expelled from the phase-separated ionogel (i.e. **T < UCST**) when held at room temperature for 4 days. Consequently, the solvent trapping degree (D_{st}) can be define as follow:

$$D_{st} = 1 - \frac{\text{Solvent loss}}{\text{Total solvent content in ionogel}} = 1 - \frac{7\%}{75\%} = 90.7\%$$

It is worth noting that though the frozen of polymer-dense phase can keep the volume unchanged of ionogel in macroscale, the solvent at the superficial zone of ionogel will be inevitably expelled from the ionogel. Therefore, it can be concluded that ionogel with smaller specific surface area will exhibit better solvent trapping capability (when held at phase-separated state). The solvent trapping capability of phase-separated ionogel was addressed in the revised manuscript and the Figure R1 was added in the Supplementary Information as Figure S6.

Figure R1. Weight retention of phase-separated ionogel as a function of time.

Comment 4. Lines 172-183: The authors discuss that the polarity of ions may not be enough to describe the behavior of ILs, and thus, suggest that the “overall properties” of ILs must be considered. While it is understandable the polarity may not be enough, but “overall properties” is still vague. Can the authors elaborate on what they mean by that term and how this needs to be considered for an IL? And, how do they support their suggestion?

Response: We thank the Reviewer for this comment. We supposed that the “overall properties” here mainly includes polarity, Lewis basicity (donor strength) and effective ionic concentration (C_{eff}). The polarity of ILs appears to be largely cation controlled, while the Lewis basicity (or donor strength) is mainly anion dependent. As a measure of the electrostatic interaction of the ILs, the effective ionic concentration (C_{eff}) is a dominant parameter for the electrostatic forces of the ILs (*J. Phys. Chem. B* **2006**, 110, 19593-19600).

Consequently, varying the cation and anion will not only affect the polarity and Lewis basicity, but also the C_{eff} of ILs.

For a certain type of anion, increasing the alkyl substituent length on cation reduces the polarity of ILs, and the C_{eff} of ILs decreases as well. In this case (i.e. increasing the alkyl chain length), the influencing trend of polarity and C_{eff} on the UCST of ionogel is the same. Consequently, it is easy to draw the conclusion that how cation side chain length affects the UCST of ionogel.

In contrast, for [C₄MIM]-based ILs with different anionic structures, C_{eff} follows the order [BF₄] > [PF₆] > [OTf] > [NTf₂] (*J. Phys. Chem. B* **2006**, 110, 19593-19600). While, the Lewis basicity of ILs that mainly determines the mutual solubility between polymer and ILs follows the order [OTf] > [BF₄] > [NTf₂] > [PF₆]. These two factors have different influencing trends on the ionogel system. However, the UCST of ionogel follows the order [BF₄] > [PF₆] > [NTf₂] > [OTf], which is different from either rule of the above-mentioned factors that caused by anion. Consequently, we hypothesize that the factor determines the UCST of ionogels is the overall properties of ILs, rather than a certain property that generated from the cation or anion in ILs.

Comment 5. Lines 271-272: Did the authors follow a specific protocol/procedure to “eliminate the interfacial gap”? was it a spontaneous step or the authors had to wait for a certain time to let the trapped gas escape?

Response: Actually, we do need to follow a specific procedure to eliminate the interfacial gaps between the ionogel and the substrate. Because the soft ionogel cannot fill the microscale roughness by its own gravity. Therefore, a certain pressure (2 ~ 5 kPa) is required to ensure the fully contact of ionogel with the substrate (Figure 4d). The reshaped surface microstructure of ionogel is fixed by cooling and a highly interlocked structure at the interface is formed. Though the step of eliminating the interfacial gaps is not entirely spontaneous, this process does not need to wait for a long time (approximately 2 ~ 3 minutes, depending on the cooling rate).

Comment 6. Figure 1: what do “opaque loose elastic” black symbols show on the Figure 1b (left)? Are they missing from the figure or they must be in grey?

Response: We thank the Reviewer for this comment. These words (opaque, loose and elastic) describe the state of the polymer-dense phase in phase-separated ionogel (Figure 1b left). Consequently, these black symbols here should be in gray according to the Reviewer’s suggestion. We have revised the Figure 1b in the manuscript.

Comment 7. Typos and grammatical errors throughout the manuscript. Examples: lines 32-34, line 63 (“research” is not countable), lines 184-186, line 189 (“effect” is a noun and “affect” is a verb), lines 213-216, lines 217-219, lines 246-247.

Response: We thank the Reviewer for these suggestions. We have carefully re-read the whole manuscript and polished the language at places by correcting several typos and misleading expressions (highlighted in the revised manuscript). The revised sentences are noted below:

(1) For example, combined with low melting point alloy, hydrated salt or crystalline polymer, the stiffness-changing ratio of the polymer composites can reach 10^4 - 10^5 . While, these binary systems depend solely on the intrinsic properties of phase change components, and thus suffer from limited tunability.

(2) The effects of cation and anion on the mechanical property of PNIPAm ionogels are characterized by comparing the elastic modulus of PNIPAm ionogels in different ILs solvents.

(3) Besides, it is found that the wider the phase separation curve opens, the higher it intersects with the T_g curves. The difference in the opening size of phase separation curves in Fig. 3e may be generated from the difference in phase separation kinetics of PNIPAm ionogel. As mentioned above, the phase separation in $[C_5MIM][PF_6]$ -ionogels possesses fast kinetics, meaning that the right part of the phase separation curve is relatively less steep, so that the Berghmans’ point emerged at an elevated temperature.

(4) For the applications of ionogels in a given area such as wearable devices, thermoresponsive ionogel within body temperature ($25^\circ\text{C} \sim 37^\circ\text{C}$) can be prepared by regulating the feed ratios of constituents.

Reviewer #3:

Comment 1. In this paper, the authors developed a material that show rubbery-to-glassy transition upon cooling based on a binary polymer-solution system with an upper critical solution temperature (UCST) and a high T_g of the polymer. Similar rubbery-to-glassy transition upon heating based on a binary polymer-solution system displaying a lower critical solution temperature (LCST) with a high T_g of the polymer has been reported recently (Nonoyama et al., *Advanced Materials* 2020). Although this work reports a thermal softening while the previous report by Nonoyama et al is thermal stiffening, the underlying physical concept are the same, that is, using liquid-liquid phase separation of binary polymer-solution system and polymer vitrification to achieve large modulus switching, which largely weakens the novelty of this work. However, since few liquid-liquid phase separation systems have the binodal curve and the glass transition curve intersected (so called Berghmans' point) in the observation temperature window, this work using PNIPAm and ionic liquid is valuable in extending the material design range.

Response: We greatly thank the Reviewer's time and effort for evaluating our manuscript as well as the very positive and constructive comments.

Comment 2. The previous paper (Nanoyama et al, *Advanced Materials* 2020) should be properly cited.

*Response: We appreciate the Reviewer's helpful suggestion. This representative work (Nanoyama et al, *Advanced Materials* 2020) has been cited in the supporting information. In the revised manuscript, this paper is also cited properly in the main text.*

Comment 3. Since the Berghmans' point freeze the kinetics, the temperature-transmittance curves (figure S3) should have a strong cooling rate dependence when UCST is close or lower than T_g . Therefore, the binary curve in Figure 2b and UCST curves in Figure 3b, d, e, f for the related data points should also strongly depend on the cooling rate. The authors should carefully address such cooling rate effect. I am wondering how the authors could determine the UCST when it is lower than T_g .

Response: We appreciate the Reviewer's helpful suggestion. As mentioned by the Reviewer, the temperature-transmittance curves have a strong cooling rate dependence when UCST is close or lower than T_g . Taking 30 wt% PNIPAm/[C₁MIM][NTf₂] ionogel as an example, whose UCST is slightly lower than its T_g according to the results in Figure 2b. As shown in Figure R2, when treating the ionogel with different cooling rate from 50°C to 20°C, the UCST of ionogel shows a strong cooling rate dependence (37.5°C and 29°C for 2 K/min and 10 K/min, respectively).

This is because the T_g of polymers can be affected by the temperature-changing rate. Generally, a faster cooling rate will lead to a higher T_g . Consequently, for ionogel with similar T_g and UCST, a faster cooling rate will lead to the frozen of polymer-dense phase prior to the equilibrium composition defined by the phase separation curve. As a result, a faster cooling rate will result in more transparent ionogel as shown in Figure 2b and 2c. The cooling rate effect was addressed in the revised manuscript and the Figure R2 was added in the Supplementary Information as Figure S5.

For UCST lower than T_g , the UCST can be determined only if it is slightly lower than the T_g of ionogel. For this purpose, we utilize a slow heating/cooling rate (≤ 2 K/min), so that the ionogel has enough time to reach the equilibrium.

Figure R2. a) Temperature dependence of transmittance at 658 nm for PNIPAm ionogels under different cooling rate. We define the cloud point (i.e. UCST) values as the temperatures at which the transmittance below 90%. b) and c) Photographs showing ionogels treated with different cooling rates.

Reviewers' Comments:

Reviewer #2:

Remarks to the Author:

The authors have appropriately addressed comments from all reviewers. I now recommend the manuscript for publication.

Reviewer #3:

Remarks to the Author:

The authors addressed the questions raised by the reviewer and revised the paper properly. The paper is recommended to publish in the current form.